# Assessing routine health information system performance during the tenth outbreak of Ebola virus disease (2018–2020) in the Democratic Republic of the Congo: A qualitative study in North Kivu

**Gabriel Kalombe Kyomba**[1]*, **Guillaume Mbela Kiyombo**[1], **Karen A. Grépin**[2], **Serge Manitu Mayaka**[1], **Thérèse Nyangi-Mondo Mambu**[1], **Celestin Hategeka**[3], **Mala Ali Mapatano**[1], **Lys Alcayna-Stevens**[4], **Serge Kule Kapanga**[5], **Joël Nkima-Numbi Konde**[1], **Dosithée Bebe Ngo**[1], **Pélagie Diambalula Babakazo**[1], **Eric Musalu Mafuta**[1], **Aimée Mampasi Lulebo**[1], **Hinda Ruton**[6], **Michael R. Law**[6]

**1** Kinshasa School of Public Health, Université de Kinshasa, Kinshasa, The Democratic Republic of Congo, **2** School of Public Health, University of Hong Kong, Pokfulam, Hong Kong SAR, **3** Department of Global Health and Population, Harvard TH Chan School of Public Health, Boston, Massachusetts, United States of America, **4** Department of Anthropology, Harvard University, Cambridge, Massachusetts, United States of America, **5** Département d'Anthropologie, Faculté des Sciences Sociales, Politiques et Administratives, Université de Kinshasa, Kinshasa, The Democratic Republic of Congo, **6** Centre for Health Services and Policy Research, The University of British Columbia, Vancouver, British Columbia, Canada

\* gabykyomba@gmail.com

**Data Availability Statement:** The data supporting the findings of this paper are stored in Qualitative

## Abstract

The Democratic Republic of Congo has implemented reforms to its national routine health information system (RHIS) to improve timeliness, completeness, and use of quality data. However, outbreaks can undermine efforts to strengthen it. We assessed the functioning of the RHIS during the 2018–2020 outbreak of Ebola Virus Disease (EVD) to identify opportunities for future development. We conducted a qualitative study in North Kivu, from March to May 2020. Semi-structured interviews were conducted with 34 key informants purposively selected from among the personnel involved in the production of RHIS data. The topics discussed included RHIS functioning, tools, compilation, validation, quality, sharing, and the use of data. Audio recordings were transcribed verbatim and thematic analysis was used to study the interviewees' lived experience. The RHIS retained its structure, tools, and flow during the outbreak. The need for other types of data to inform the EVD response created other parallel systems to the RHIS. This included data from Ebola treatment centers, vaccination against Ebola, points of entry surveillance, and safe and dignified burial. The informants indicated that the availability of weekly surveillance data had improved, while timeliness and quality of monthly RHIS reporting declined. The compilation of data was late and validation meetings were irregular. The upsurge of patients following the implementation of the free care policy, the departure of healthcare workers for better-paid jobs, and the high prioritization of the outbreak response over routine activities led to RHIS disruptions. Delays in decision-making were one of the consequences of the decline in data timeliness.

Date Repository (QDR) Main Collection. They can be accessed at https://data.qdr.syr.edu/dataset.xhtml?persistentId=doi:10.5064/F6M4MJRL.

**Funding:** This paper was partially supported by a research grant from the International Development Research Centre (IDRC), Grant number, 108966-002, KG and SM. However, the specific data used in this paper were not directly linked to the original project and were instead collected as part of GK's unfunded doctoral research programme. The original funders had no role in study design, data collection and analysis, decision to publish, or preparation of the manuscript.

**Competing interests:** The authors have declared that no competing interests exist.

Adequate allocation of human resources, equitable salary policy, coordination, and integration of the response with local structures are necessary to ensure optimal functioning of the RHIS during an outbreak. Future research should assess the scale of data quality changes during outbreaks.

# 1. Introduction

## 1.1 Background

According to the World Health Organization (WHO), Health Information Systems, including Routine Health Information Systems (RHIS), are one of the six components of a health system [1]. A RHIS is an organized set of structures, institutions, procedures, methods, materials, and people for generating data [2]. The RHIS ensures the production, analysis, dissemination, and use of health information; it serves as a basis for assessing the health status of populations, the adequate functioning of the health system, and guides health service managers in making decisions concerning policy reorientation or resource allocation [3]. The RHIS is therefore a crucial component from which the other health system building blocks draw evidence to optimize their function [4].

In 2008, WHO encouraged health service reforms to better consider the needs and expectations of beneficiaries [5], notably through evidence generated by the information systems [6, 7]. More than a decade later, and despite the opportunities offered by new information and communication technologies [8], progress in implementing robust RHISs remains insufficient in the majority of low-income countries and has lagged behind the progress observed in wealthier countries [9, 10]. However, they lag not simply because of technical problems: organizational and behavioral problems have also contributed to their slow progress [3, 11].

The Democratic Republic of Congo (DRC) established its RHIS in 1987 [12]. Since then, it has undertaken several reforms [13], including the transition to the second generation of the District Health Information System (DHIS2) as the online platform for storing, sharing, and analyzing data in its RHIS [14, 15]. With the support of its partners, the deployment of DHIS2 began in 2015 and reached coverage of all districts by the end of 2016 [15]. Data are collected at facilities using paper forms, which are then transmitted to the District office where they are entered into the DHIS2 [6].

For the RHIS to function correctly, tools—such as patient records, registers, reporting forms, data entry platforms—need to be designed, updated, reproduced, and deployed; training needs to be organized and technologies such as computers and software need to be distributed to all levels of the health system [15]. Despite these efforts over several years, the DRC's RHIS still faces challenges including poor timeliness, low completeness, and poor consistency [15, 16]. Furthermore, the analysis and uptake of RHIS data by decision-makers remains suboptimal [17, 18]. Evaluations before the shift to DHIS2 did not note any significant RHIS improvements between 2009 and 2015, particularly with regards to the reporting of data in real-time [12].

The occurrence of an infectious disease outbreaks has severe consequences on many health system components. This has been the case with the Severe Acute Respiratory Syndrome (SARS) in Canada [19] and Taiwan [20], as well as with the Ebola Virus Disease (EVD) outbreak in West Africa [21–26]. However, while the delivery of health services component is largely studied, less is known about the impact of infectious disease outbreaks on the functioning of the health information system.

From 2018 to 2020, in the provinces of North Kivu, Ituri, and South Kivu, the DRC recorded its tenth EVD outbreak. This outbreak remains the most serious in the country in terms of its duration, the size of the affected area, and the number of victims [27]. The regions affected by this outbreak were victims of the challenges that also hamper the proper RHIS functioning in the rest of the country. These include geographical inaccessibility, poor mobile phone and internet coverage [17, 27, 28], as well as insufficient resources, particularly human resources [29, 30]. In addition, there are ongoing security issues in these regions [27]. The combination of these factors is likely to undermine efforts to improve RHIS performance.

Unlike the other components of the health system, the performance of the RHIS is seldom evaluated in the DRC and even more rarely during outbreaks of infectious diseases. For example, Lal et al. have focused on RHIS' areas such as governance and coordination, infrastructure and resources of health systems during the Ebola and Covid-19 epidemics [31]. Moreover, in the DRC, as elsewhere, RHIS evaluations in non-epidemic contexts have largely used quantitative indicators to assess performance [9, 12, 32, 33]. These studies have focused more on RHIS output rather than on the functioning of the data production process. The present study was therefore designed to fill the evidence gap on RHIS functioning during an EVD outbreak in general and during the North Kivu EVD outbreak in particular. Ultimately, this study aimed to contribute to the strengthening of the RHIS in future public health crises by generating lessons on what did or did not work. It is in line with WHO's vision of encouraging countries to build health systems that are more resilient to shocks–including disease outbreaks [34].

## 1.2 Conceptual framework

We used the Performance of Routine Information System Management (PRISM) conceptual framework developed by Measure Evaluation. PRISM makes it possible to evaluate RHIS components and make comparisons between countries and across time [3, 11]. Its application has helped to identify weaknesses and to develop plans to strengthen national RHIS in other international contexts [11]. For example, several countries in Africa have benefited from a PRISM assessment [12].

As shown in Fig 1, the PRISM framework includes three determinants of RHIS performance: technical, organizational, and behavioral aspects. According to Measure Evaluation [12] and Aqil et al. [11], PRISM is based on the following three assumptions: (i) a well-functioning RHIS produces and uses quality data; (ii) the performance of the RHIS depends on the environment, available resources, practices, and processes; and (iii) decisions based on quality data lead to a strengthened health system, which improves the health status of the population. In this study, our focus was on inputs, processes, and output of the PRISM framework.

## 2. Materials and methods

### 2.1. Ethics statement

This research was approved by the Ethics Committee of the Kinshasa School of Public Health (No ESP/CE/11/2020). Participation in the study was free, voluntary, without financial compensation and without individual benefit. The potential risks that were presented to participants in order to obtain their verbal consent were the loss of time spent on the interview and possible reprisals against their criticism or opinion. The verbal consent was also sought for the recording of the interview on a Dictaphone. Participants were informed of the confidentiality and anonymity measures in place. Indeed, the interviews were one on one and conducted in privacy. Finally, the results are presented in a global and anonymous way to prevent potential identification of participants. One informant did not wish to be recorded. He was not included in the study.

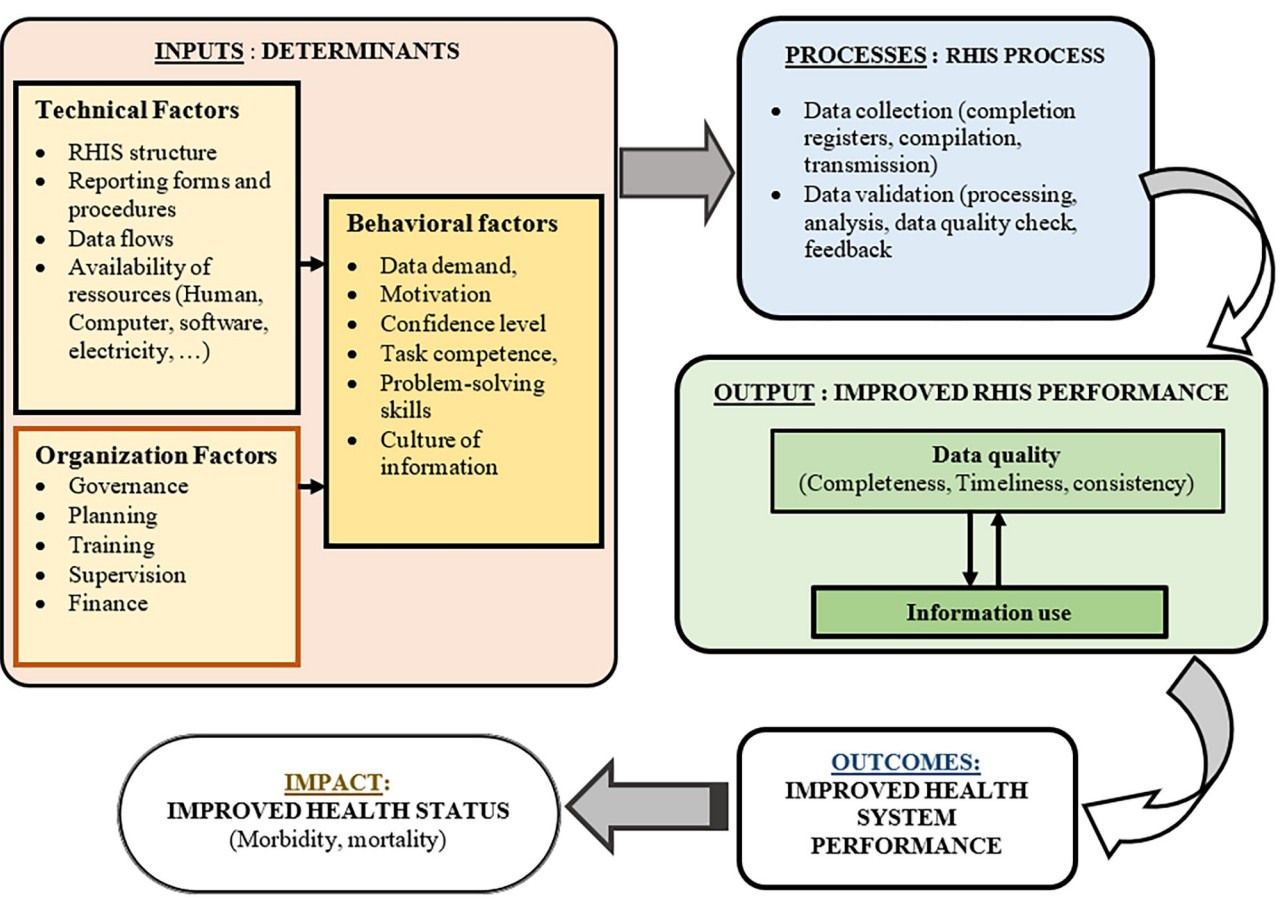

**Fig 1. PRISM conceptual framework.**

## 2.2. Type, site, and period of study

We conducted a qualitative case study of the performance of the RHIS during the 2018 to 2020 EVD outbreak in the Province of North Kivu, Eastern DRC. We selected 5 of 19 affected districts: as shown in Fig 2. In order to assess the upper limit of the effect on RHIS, the districts selected were those where the severity of the epidemic was higher relative to the total number of EVD cases reported throughout the epidemic.

The interviews were conducted between March 9th and May 14th 2020. This period was part of the tail end of the outbreak, which was first declared on August 2nd 2018 and was officially declared over on June 25th 2020. However, all response interventions were still in place at the time of data collection. This includes case management, surveillance, contact tracing, vaccination, and free care in most healthcare facilities. This period also corresponds to the first cases of COVID-19 in the country. However, the study site was not heavily affected by this new pandemic at that time.

## 2.3. Sampling

**Profile of respondents.** We recruited study participants from all staff involved in the production and use of health data at the provincial and operational levels. Key informants (KIs) from the Provincial Health Department were recruited from the office managers, analysts, and program managers. Representatives of non-governmental organizations (NGOs) that provided

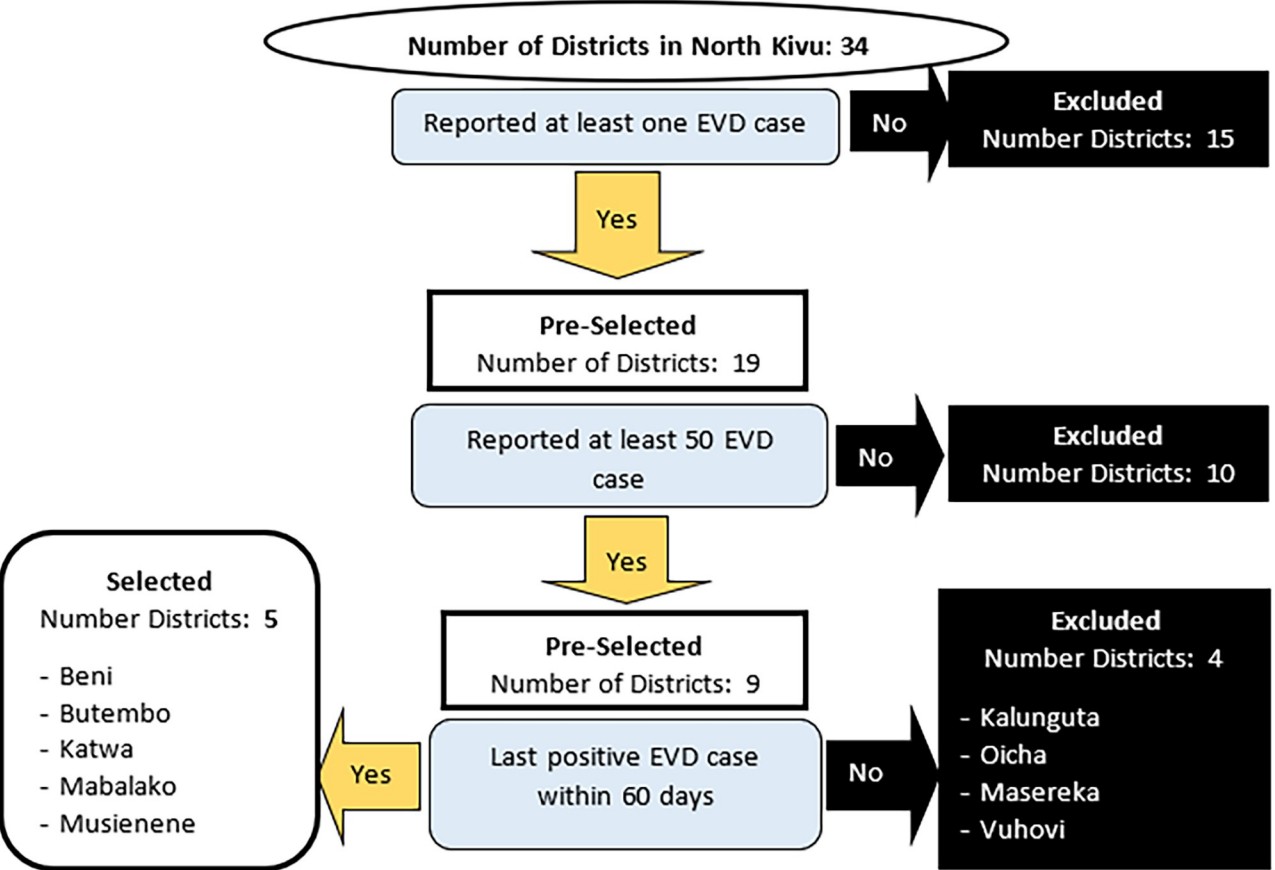

**Fig 2. Selection scheme for the health districts included in the study.**

financial and technical support for routine health services were also targeted. At the operational level, KIs were selected from the health District's office and the facilities. KIs from districts were physicians, administrators, or supervising nurses. In facilities, KIs were selected among nurses in charge of the health area and members of the hospital management committees, such as medical directors, chief of staff physicians, nursing directors, and administrators.

**Sample size and KIs selection.** We interviewed 34 KIs. Even though we achieved saturation (this was reflected in the repetition of the same information and the absence of new ideas) with about 20 semi-structured interviews (SSIs), we continued interviewing other informants to have a better representation of several sub-samples such as the duration of the outbreak, the number of reported cases of EVD in the region, as well as the profile of the KIs. We also considered other events that may have positively or negatively affected the health services during the period under study. For example, it has been shown that the free healthcare policy (partial or total) had an important effect on the use of services was documented during the 2018 EVD outbreak in Equateur province [35]. In addition to this free care policy that was also implemented, the presence or absence of insecurity, attacks, or threats strongly characterized the North Kivu province during the period under study.

*Selection of districts, facilities, and KIs.* Districts that had reported at least one confirmed case of EVD were listed. From this list, we retained only the districts that had reported more than 50 confirmed cases of EVD and whose last case was not older than two months at the time of selection (see Fig 2). To reduce subjectivity in the selection of facilities in each District,

we listed *the three* health areas: *i) to first experience the outbreak*, *ii) with the highest number of EVD cases* and *iii) that experienced the most attacks or resistance*. Health areas on several lists, as well as those at the top of each list, were retained. Reasoned choice was also used to select facilities and KIs at different levels. This non-probability sampling method ensures the identification and the inclusion of informants who have more information on the research.

Inclusion and exclusion criteria: the selected districts had to have reported 50 or more EVD cases and had their last case within the past 60 days. As for respondents, selected KIs were required to have been in their current role for at least one year before the EVD outbreak, as well as throughout the outbreak. They were also required to be present, available, and willing to participate in the study. Containment measures introduced as part of the fight against Covid-19 limited access to certain health facilities.

## 2.4. Data collection

Data were collected through semi-structured interviews (SSIs) conducted with KIs. A pretested interview guide based on the PRISM framework was used. All SSIs were conducted either in French or Swahili by the first author who is a PhD student. Nearly all interviews were one-on-one, were conducted at the respondents' place of work, and in private. One interview was conducted via telephone and one outside of regular office hours and location due to the availability of the respondent. Audio recordings were taken on a Dictaphone, as well as notes, including non-verbal expressions, were also taken during each interview, which on average lasted 50 minutes.

## 2.5. Data quality assurance

All interviews were conducted by the first author, which avoided the problem of delegation of tasks inherent to qualitative research. The triangulation of sources was chosen by the research team to ensure the quality of the data and therefore the validity of the results. This included the selection of KIs from several professional categories (doctors vs nurses vs administrators), functions (providers vs managers), levels in the health system pyramid (Health facilities vs districts vs provincial), and affiliation (governmental actors vs supporting partners). Finally, sharing and discussing the preliminary results with the management team and with the actors in the sector helped to minimize misinterpretations.

## 2.6. Data analysis

The audio recordings in French were transcribed verbatim, while those in Swahili were translated to French during transcription by two members of the research team who are fluent in both languages. The transcripts were read several times by the team alongside the audio recordings. This allowed for the accuracy of the transcription and translation to be assessed. It also allowed for familiarization with the content of the transcripts.

After cleaning, the transcripts were imported to Atlas-TI version 7.5.7 for coding. The analysis plan was developed using transcripts from the first five interview transcripts and core concepts of the PRISM framework. Thematic analysis was performed to understand participants' lived experiences. This qualitative approach proceeds systematically by identifying, grouping, and examining expressions that emerge in a corpus and are related to the themes under investigation [36]. All the themes from KIs' narratives were about different dimensions of the PRISM framework (See Coding tree, S1 Fig).

## 3. Results

### 3.1. Sample description

The study sample comprised 34 KIs selected from the provincial health department (n = 7) and the districts of Beni (n = 10), Butembo (n = 7), Katwa (n = 6), Mabalako (n = 3), and ZS Musienene (n = 1). The socio-demographic and occupational characteristics of KIs are described in Table 1.

As shown in Table 1, the majority of study participants were men (79%), nurses (59%), and had attended university (79%). Half of them worked in the facilities from which the data originated. This part included 11 (32%) nurses in charge of the health areas and 6 (18%) hospital managers. There were also 7 (20%) supervisors or analysts, 5 (15%) program managers or office managers, and 5 (15%) administrators. In this sample, the average age of the participants was 47.7 years (range ±6.3 years), while the average length of time in the job was 18.8 years (range ±8.2 years).

### 3.2. RHIS structure, forms, and data flow during the outbreak

Respondents indicated that facilities continued to produce and transmit the same types of data using the usual forms and channels according to the set deadlines during the outbreak. The KIs did not report any changes in the guidelines for the frequency and deadlines of reporting. However, the KIs reported that there was a need for other types of data for the various response commissions and their partners. These include data related to case management in

**Table 1. Characteristics of the key informants.**

| Socio-demographic characteristics | PHD* | District | Hospital | Health centre | Total |
|---|---|---|---|---|---|
| | n = 7 | n = 10 | n = 6 | n = 11 | n (%) = 34 |
| **Gender** | | | | | |
| Male | 6 | 7 | 6 | 8 | **27 (79.4)** |
| Female | 1 | 3 | 0 | 3 | **7 (20.6)** |
| **Age category (years)** | | | | | |
| 35–40 | 1 | 2 | 0 | 0 | **3 (8.8)** |
| 41–54 | 4 | 6 | 4 | 8 | **22 (64.7)** |
| 55 + | 0 | 0 | 1 | 1 | **2 (5.9)** |
| Not specified | 2 | 2 | 1 | 2 | **7 (20.6)** |
| **Years of experience** | | | | | |
| 3–10 | 2 | 1 | 0 | 1 | **4 (11.8)** |
| 11–20 | 2 | 5 | 5 | 6 | **18 (52.9)** |
| 21 + | 0 | 3 | 1 | 4 | **8 (23.5)** |
| Not specified | 3 | 1 | 0 | 0 | **4 (11.8)** |
| **Qualification/Function** | | | | | |
| Nurses | 2 | 5 | 2 | 11 | **20 (58.8)** |
| Administrators | 2 | 3 | 2 | 0 | **7 (20.6)** |
| Physician | 3 | 2 | 2 | 0 | **7 (20.6)** |
| **Education level** | | | | | |
| Secondary | 0 | 0 | 0 | 3 | **3 (8.8)** |
| University | 4 | 9 | 6 | 8 | **27 (79.4)** |
| Post University | 3 | 1 | 0 | 0 | **4 (11.8)** |

PHD: *Provincial Health Department*,

* including Non-Governmental Organization Staff.

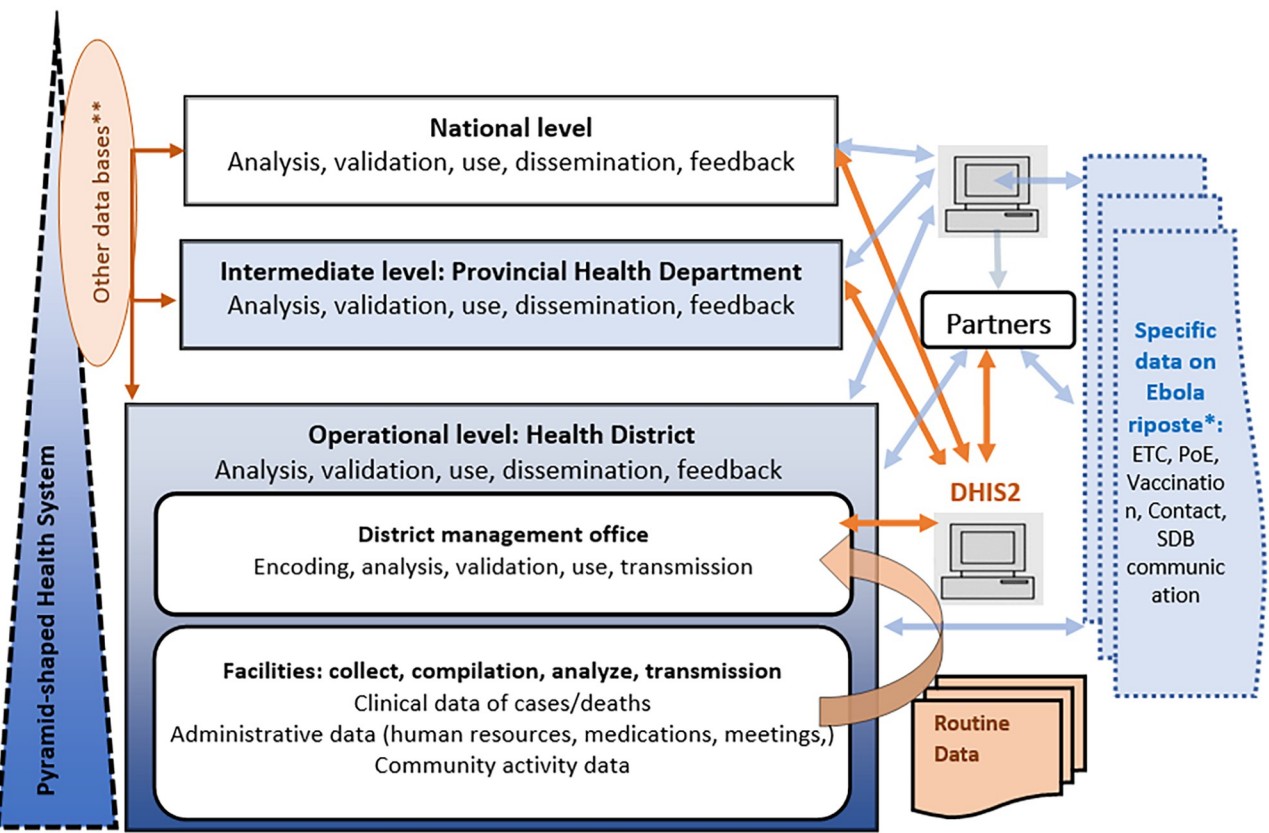

**Fig 3. Data flow during the 2018–2020 EVD outbreak, North Kivu Province, Eastern DRC.** DHIS2: *second generation of the District Health Information System.* *ETC: *Ebola treatment center*, PoE: *point of entry*, SDB: *safe and dignified burials.* ** *Weekly surveillance data, data from Specific programs*

Ebola treatment centers, vaccination against Ebola, contact surveillance, point of entry and community level alerts, and dignified and safe burials. As described in Fig 3, these data were captured through parallel channels which were not interconnected with the DHIS2.

## 3.3. Overall assessment of the functioning of the RHIS

Two trends emerged regarding the functioning of the RHIS during the EVD outbreak. A minority felt that the functioning was normal or almost normal. However, this minority came mainly from hospitals and first-line facilities that reported fewer cases of EVD or whose staffing had remained consistent throughout the outbreak. In contrast, the majority of KIs felt that the RHIS had been disrupted at the facility, district, and provincial levels. They mentioned that disruptions were most severe at the beginning of the outbreak and during three weeks following the notification of an EVD positive case. The following comment illustrates this idea:

"*When we had a positive case, we were really sick, for all of twenty-one days because we were not stable (. . .) We didn't have time to put ourselves and think about anything other than that point of a positive Ebola case.*"

[Nurse, Facility / Code: IA_01_AS_IT_01]

### 3.4. RHIS process

**Completion of registers and data compilation.**   The majority of KIs felt that the completion of registers was not affected because they were the source of verification for the purchase of services under the performance-based financing system and the basis for active epidemiological surveillance. It was therefore in the interest of the facilities to complete them regularly. On the other hand, a minority indicated that the registers were sometimes incomplete or completed late when data compilation took too long. According to them, the large flow of patients after implementation of the free healthcare policy had increased the providers' workload. Below is a facility nurse's quote:

"*I told you earlier that at the beginning, we were receiving, for example, 200 patients per month. But today we have 200 patients per day. The time to record all these patients in the register, you don't have that time.*"

[Nurse, Facility / Code: IA_01_AS_IT_02]

**Data validation and use for decision-making.**   The vast majority of KIs reported that data analysis suffered greatly during the EVD outbreak, sometimes resulting in inconsistent estimates. Informants reported that data validation meetings were held infrequently, with many postponements. Quorum problems, delays in receiving reports, and the prioritization of the response over routine activities emerged as causes for the non-scheduling of routine meetings. This resulted in a delay in decision-making. In contrast, a few KIs indicated that data validation meetings had continued normally. Due to the scarcity of supervision visits or data validation meetings, feedback was lacking. When it was given, the expected actions were not always carried out.

Inadequate analysis of data had consequences, including the late detection of other epidemics. Few informants reported that measles outbreak was identified late. Thus, the health system was not able to implement strategies that can improve health. All this required data availability, analyses, and holding meetings, as illustrated by this quote:

"*During the Ebola outbreak we also had the measles outbreak but we didn't notice because everyone was focused on Ebola. It was only later that we realized that the measles outbreak was there, and even a cholera outbreak. We were slapped with deaths that were already coming and we were all focused on Ebola*". . ."

[Nurse, Provincial, / Code: IA_03_DPS_Analyst_01]

### 3.5. KI's perceptions of RHIS performance

**Timeless and completeness.**   The vast majority of KIs reported that data transmission was significantly delayed. However, they acknowledged that data completeness was not an issue. In contrast, despite high levels of report completeness, a minority of informants felt that there was missing data in the templates. One respondent mentioned the loss of data on non-Ebola diseases treated in Ebola treatment centers. Indeed, after their investigation, when the patient's condition improved and the Ebola test was negative, patients left the centers for home, resulting in under-reporting of their diseases. Another respondent noted that some private facilities had stopped reporting or were reporting after insistence from health officials, in protest at the fact they were not covered by financial aids as public facilities. For example, one nurse indicated that:

"*I would say that at home, it did not cause any disruption. Except, where it caused a small disruption, it was at the level of the private hospitals that are in our health areas. We had difficulties collecting their data during the period of the outbreak.*"

[Nurse, Facility / Code: IA_04_AS_IT_02]

**Data consistency or accuracy.**　The majority of our KIs felt that the quality of health data was affected during the EVD outbreak. Some KIs felt that unintentional errors were likely to occur in the templates because of the very high volumes of data to be compiled and the pressure on providers to send in their reports within the deadline. A minority felt that some numbers were filled in on the templates without any underpinning because providers were avoiding sanctions from their superiors, as illustrated by this response:

"*. . . First of all, I can say that there was fabricated data. So, when they felt they were late in making the report, they could put in random numbers. (. . .) So he made the outline in a hurry to satisfy us.*"

[Supervisor, District / Code: IA_05_BCZS_IS]

## 3.6. Challenges and initiatives to maintain RHIS optimal functioning

The RHIS continues to face several additional challenges not specific to the EVD outbreak. These include geographical inaccessibility and poor internet coverage. Prior to the epidemic, facilities were sometimes the targets of threats from armed groups active in the region. Some facilities had even been burned or looted, and some health providers had been killed or kidnapped. In addition to these attacks, which continued or worsened during the outbreak, unhappy community members also attacked facilities, providers and supporting partners. Some informants explained that these attacks had been motivated by the fact that some providers had been found to be corrupt and complicit with those who instigated the EVD outbreak for their hidden interests. This environment of fear hindered work in general and data reporting activities specifically.

Moreover, one informant reported the destruction of registers and patient follow-up records of human immunodeficiency virus service by the disinfection team. This radical solution resulted in a huge loss of records for a system that is not digitized. Below is a quote from KIs to support this observation:

"*Unfortunately, the voluntary testing center* [for human immunodeficiency virus -HIV-] *was next to where the first confirmed cases were registered. So, when they came to decontaminate the rooms where were these patients, they burnt all our files. All the files of our patients living with HIV infection. To reconstitute the files of all these patients, it gave us a lot of difficulties.*"

[Supervisor, District / Code: IA_01_BCZS_IS]

Other respondents admitted that the departure of experienced staff also affected data reporting. To cope with this challenge, some facilities recruited agents responsible for collecting data with the support of NGO partners. Other facilities opted for teamwork, the delegation of tasks, or recruitment of staff, even if the necessary skills were sometimes lacking.

In expressing their wishes for the future, most respondents felt that the addition of staff or the establishment of an equitable salary policy between outbreak agents and routine health

service providers should be considered. They explained that while workload reduction requires adding staff, fair treatment of providers is likely to encourage staff to stay in one facility for a longer period. To illustrate this, one managing director said:

> ". . . *we will have to put the providers on the same level first, the providers who work in the outbreak activities and those who will remain doing other routine activities in the facilities, they will have to have the same motivation so that the disruption is not visible in this community.*"

[Administrator, Hospital / Code: IA_01_HGR_AG]

## 4. Discussion

This study aimed to assess the RHIS performance during the EVD outbreak in North Kivu. We found that the RHIS had continued to function, notably because the data were still produced. The need to have sources of verification for disease surveillance and the purchase of services under the PBF and the fear of administrative sanctions were among the reasons for maintaining the RHIS during the outbreak. Other factors were previously identified by Chanyalew et al as predictors of data use in the Amhara Region, Ethiopia. These include the existence of the instructions, the need for use of data for target setting, and the culture of displaying performance data [37].

The study highlighted that the flow of health information changed with the creation of parallel data systems which were not integrated into the DHIS2. Indeed, the various commissions of the outbreak response and their partners needed additional information that was not included in the usual reporting forms. This multiplicity of reports had increased the volume of work and could lower the data quality [28]. Reducing the number and simplifying data reporting forms, as tested in Mozambique, may be an option to consider during an outbreak [8]. In addition, the complexity of the data flow and the multiplicity of databases may make it more complicated to analyze the data. This can hinder decision-making, especially as some stakeholders reported that they did not have access to all these databases.

The influx of patients following the introduction of free health care has led to an increase in the volume of work. At the same time, the number of staff decreased as many providers joined the response team. As a result, filling out registers and compiling data had become very laborious, leading to the decline in the timeliness of reporting. This likely moved the country away from its goal of migrating to real-time data reporting [28].

This study revealed the difficulty of maintaining regular data analysis, validation meetings, supervision, and feedback during an outbreak. The challenge of achieving a quorum and the poor timeliness of reports had made some meetings difficult to complete. As a result, a measles outbreak was identified late. This finding would be in line with the findings of Wickremasinghe et al. who reported that data availability, data quality, and human dynamics were among the barriers to data-driven decision making [8]. Moreover, given their importance on data quality as reported by Ahanhanzo et al. [29], or on the use of data at the peripheral level as reported by Chiferaw et al. [38], a focus on feedback and supervision could be beneficial to the RHIS. This is particularly important because the DRC RHIS outside of the EVD outbreak context is already characterized by insufficient analysis and low data utilization [13, 15, 18].

Informants thought that there was a decline in data accuracy during the EVD outbreak. It was reported that some templates were submitted with missing or fabricated data when it was difficult to respect the deadline as data compilation was laborious. This practice had the potential to increase discrepancies between the primary sources and the data encoded in the DHIS2.

As a result, the problem of over- and under-reporting in facilities affected by the EVD outbreak under study may be more acute than those observed to varying degrees in Rwanda [33], Ethiopia [32, 39], Gombe in Nigeria [40], in South Africa [41], or the West Bank in Palestine [42]. Others assessments of RHIS outside the outbreak have revealed problems with data quality [43]. In Benin, Ahanhanzo et al. [30] reported a significant association between data quality and the availability of resources. Venkateswaran et al. [42] reported that discrepancies between source documents and reports. All these factors and effects are likely to be magnified during the EVD outbreak, which may affect RHIS performance [31]. Quantitative studies could help shed light on the existence and the magnitude of over- and under-reporting and factors associated to data quality concerns during an Ebola outbreak.

Overall, respondents' views on the occurrence of disruptions in the functioning of the RHIS during the EVD outbreak were divergent. There were no disruptions for some, while there were for others. Based on the opinions of the KIs, there was a direct relationship between the magnitude of the outbreak, particularly the number of cases of EVD, and the level of disruption of the RHIS. Form some KIs, all activities related to data reporting were affected in the same way while for others, the compilation and holding of data analysis meetings were very much affected. Several respondents reported also that the critical periods were the beginning of the outbreak and the weeks following the notification of confirmed EVD cases. In a context of limited resources, efforts can be focused in the most affected districts, on the most vulnerable activities, and during the most critical periods. Limited access to the internet and electricity, as well as insecurity, were also reported as challenges to data reporting. However, these were not specific to the EVD outbreak because they have already been reported across the country [18]. They have also been identified as a handicap to the RHIS performance in Liberia [29] and some low-income countries [8].

Several initiatives that kept the RHIS functional during the outbreak were reported. They included the delegation of tasks, the creation of data manager positions, and increased staffing levels. Although assumed to be effective by respondents, the presence of a data officer had not improved the availability and completeness of data in Malawi in a non-epidemic setting [44]. In contrast, we did not document the implementation of electronic registers in routine healthcare services despite their advantages. In fact, in some Zambian facilities, electronic patient records had led to the elimination of the compilation stage, real-time reporting, automatic generation and sending of quality reports, and the use of health information at different levels [8].

While we acknowledge that the impact of the initiatives documented here extends far beyond the scope of this study, we believe that a deeper analysis of the causes of the disruptions is warranted before the appropriate solutions to these problems is identified. [35]. In fact, Wetherill reported that the introduction of technical factors or new technologies was favoured, whereas the root causes of poor data quality were behavioural and organizational which include limited human resource capacity and the lack of a culture of data collection, communication, and use.

## 5. Study limitations and perspectives

We did not objectively assess the quality of the data produced during the EVD outbreak to validate the opinions of our KIs. This limitation offers an opportunity for future research. Similarly, it may be important to assess the effect of initiatives taken during the outbreak on the performance of the RHIS. Nonetheless, the sample size and the selection of participants at three different levels, namely the provincial, the districts, and the facilities allowed the triangulation of data sources and improved the validity of the results.

## 6. Conclusion

This study assessed the process of health data production during the tenth outbreak of EVD in North Kivu, DRC. It showed that the flow of data did not change but that the RHIS faced several challenges that could affect its performance. The integration of the outbreak response into local structures to allow for a single command unit for routine services and the response, the allocation of human resources according to workload, the implementation of a fair and incentive-based salary policy, the simplification and centralization of data flow and the implementation of electronic registers should be considered to build an efficient RHIS during future public health crises.

## Supporting information

**S1 Fig. Coding tree.**
(TIF)

## Acknowledgments

We acknowledge the contribution of Ramazani Yuma, former Secretary-General for Health, Ministry of Health, DRC for supporting the study. We also thank doctor Justin Kabondjo, Director of the health information system, and doctor Janvier Kabuya, Head of the provincial Health Department in North Kivu for their support of the study. We are grateful to all the key informants for helping us obtain information on key parameters related to this study.

## Author Contributions

**Conceptualization:** Gabriel Kalombe Kyomba, Karen A. Grépin, Serge Manitu Mayaka, Michael R. Law.

**Data curation:** Gabriel Kalombe Kyomba.

**Formal analysis:** Gabriel Kalombe Kyomba, Serge Kule Kapanga.

**Funding acquisition:** Gabriel Kalombe Kyomba, Serge Manitu Mayaka.

**Investigation:** Gabriel Kalombe Kyomba.

**Methodology:** Gabriel Kalombe Kyomba, Guillaume Mbela Kiyombo, Karen A. Grépin, Serge Manitu Mayaka, Thérèse Nyangi-Mondo Mambu, Celestin Hategeka, Mala Ali Mapatano, Lys Alcayna-Stevens, Joël Nkima-Numbi Konde, Dosithée Bebe Ngo, Pélagie Diambalula Babakazo, Eric Musalu Mafuta, Aimée Mampasi Lulebo, Michael R. Law.

**Project administration:** Gabriel Kalombe Kyomba, Serge Manitu Mayaka.

**Resources:** Gabriel Kalombe Kyomba, Karen A. Grépin, Serge Manitu Mayaka.

**Software:** Gabriel Kalombe Kyomba.

**Supervision:** Gabriel Kalombe Kyomba, Guillaume Mbela Kiyombo, Karen A. Grépin, Serge Manitu Mayaka, Thérèse Nyangi-Mondo Mambu, Joël Nkima-Numbi Konde.

**Validation:** Gabriel Kalombe Kyomba.

**Visualization:** Gabriel Kalombe Kyomba.

**Writing – original draft:** Gabriel Kalombe Kyomba.

**Writing – review & editing:** Gabriel Kalombe Kyomba, Guillaume Mbela Kiyombo, Karen A. Grépin, Serge Manitu Mayaka, Thérèse Nyangi-Mondo Mambu, Celestin Hategeka, Mala

Ali Mapatano, Lys Alcayna-Stevens, Serge Kule Kapanga, Joël Nkima-Numbi Konde, Dosithée Bebe Ngo, Pélagie Diambalula Babakazo, Eric Musalu Mafuta, Aimée Mampasi Lulebo, Hinda Ruton, Michael R. Law.

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
