## [Decision Letter · Decision Letter 0]

8 Feb 2022

PGPH-D-21-01117

Assessing Routine Health Information System Performance during the Tenth Outbreak of Ebola Virus Disease in the Democratic Republic of the Congo: a qualitative study in North Kivu

Dear Dr. Kyomba,

Thank you for submitting your manuscript to PLOS Global Public Health. After careful consideration, we feel that it has merit but does not fully meet PLOS Global Public Health’s publication criteria as it currently stands. Therefore, we invite you to submit a revised version of the manuscript that addresses the points raised during the review process.

We look forward to receiving your revised manuscript.

Kind regards,

Ari Natalia Probandari, PhD

Academic Editor

Journal Requirements:

1. Please ensure that the funders and grant numbers match between the Financial Disclosure field and the Funding Information tab in your submission form. Note that the funders must be provided in the same order in both places as well.

2. Please amend your detailed Financial Disclosure statement. This is published with the article, therefore should be completed in full sentences and contain the exact wording you wish to be published.

i). State the initials, alongside each funding source, of each author to receive each grant.

ii). State what role the funders took in the study. If the funders had no role in your study, please state: “The funders had no role in study design, data collection and analysis, decision to publish, or preparation of the manuscript.”

iii). If any authors received a salary from any of your funders, please state which authors and which funders.

3. Please update your Competing Interests statement. If you have no competing interests to declare, please state: “The authors have declared that no competing interests exist.”

4. In the online submission form, you indicated that “Data supporting the findings of this study are available from the corresponding author GKK on request.”. All PLOS journals now require all data underlying the findings described in their manuscript to be freely available to other researchers, either 1. In a public repository, 2. Within the manuscript itself, or 3. Uploaded as supplementary information.

5. Please provide separate figure files in .tif or .eps format only and ensure that all files are under our size limit of 20MB.

Additional Editor Comments (if provided):

The authors should follow COREQ guideline for qualitative research reporting. In the guideline, the authors should provide overall coding tree and give examples of coding process. The authors also should provide explanation on researchers subjectivity and reflexivity (see COREQ). In addition to that, please respond to reviewers comment and give revisions accordingly.

Reviewers' comments:

Reviewer's Responses to Questions

**Comments to the Author**

1. Does this manuscript meet PLOS Global Public Health’s publication criteria? Is the manuscript technically sound, and do the data support the conclusions? The manuscript must describe methodologically and ethically rigorous research with conclusions that are appropriately drawn based on the data presented.

Reviewer #1: Yes

Reviewer #2: Yes

2. Has the statistical analysis been performed appropriately and rigorously?

Reviewer #1: Yes

Reviewer #2: N/A

3. Have the authors made all data underlying the findings in their manuscript fully available (please refer to the Data Availability Statement at the start of the manuscript PDF file)?

Reviewer #1: Yes

Reviewer #2: Yes

4. Is the manuscript presented in an intelligible fashion and written in standard English?

Reviewer #1: Yes

Reviewer #2: Yes

5. Review Comments to the Author

Reviewer #1: Manuscript: "Assessing Routine Health Information System Performance during the Tenth Outbreak

of Ebola Virus Disease in the Democratic Republic of the Congo: a qualitative study in

North Kivu (PGPH-D-21-01117)" has been submitted to “PLOS Global Public Health” for consideration.

The manuscript can be accepted for publication in its current form..

Reviewer #2: Please see my comments in the attached file. This is a nicely executed study. I only have a number of minor, even trivial, comments, mostly meant to add clarity. If the editor decides to request a revision, please do not provide a point-by-point response to my comments. Instead, address them directly in the track-change version of the manuscript (and fine to remove my comments, as I have a separate copy saved).

6. PLOS authors have the option to publish the peer review history of their article (what does this mean?). If published, this will include your full peer review and any attached files.

**Do you want your identity to be public for this peer review?** For information about this choice, including consent withdrawal, please see our Privacy Policy.

Reviewer #1: No

Reviewer #2: No

---

## [Decision Letter · Decision Letter 1]

5 Jul 2022

Assessing Routine Health Information System Performance during the Tenth Outbreak of Ebola Virus Disease (2018- 2020) in the Democratic Republic of the Congo: a qualitative study in North Kivu

PGPH-D-21-01117R1

Dear Doctor Kyomba,

We are pleased to inform you that your manuscript 'Assessing Routine Health Information System Performance during the Tenth Outbreak of Ebola Virus Disease (2018- 2020) in the Democratic Republic of the Congo: a qualitative study in North Kivu' has been provisionally accepted for publication in PLOS Global Public Health.

Best regards,

Lucinda Shen, MSc

PLOS Staff Editor 

on behalf of 

Ari Natalia Probandari, PhD

Academic Editor

Please respond to the reviewer's comment and revise accordingly.

Reviewer Comments (if any, and for reference):

Reviewer's Responses to Questions

**Comments to the Author**

1. If the authors have adequately addressed your comments raised in a previous round of review and you feel that this manuscript is now acceptable for publication, you may indicate that here to bypass the “Comments to the Author” section, enter your conflict of interest statement in the “Confidential to Editor” section, and submit your "Accept" recommendation.

Reviewer #2: All comments have been addressed

2. Does this manuscript meet PLOS Global Public Health’s publication criteria? Is the manuscript technically sound, and do the data support the conclusions? The manuscript must describe methodologically and ethically rigorous research with conclusions that are appropriately drawn based on the data presented.

Reviewer #2: Yes

3. Has the statistical analysis been performed appropriately and rigorously?

Reviewer #2: N/A

4. Have the authors made all data underlying the findings in their manuscript fully available (please refer to the Data Availability Statement at the start of the manuscript PDF file)?

Reviewer #2: Yes

5. Is the manuscript presented in an intelligible fashion and written in standard English?

Reviewer #2: Yes

6. Review Comments to the Author

Reviewer #2: I want to thank the authors for addressing all of my comments in the previous version.

7. PLOS authors have the option to publish the peer review history of their article (what does this mean?). If published, this will include your full peer review and any attached files.

**Do you want your identity to be public for this peer review?** For information about this choice, including consent withdrawal, please see our Privacy Policy.

Reviewer #2: No
